# Laser Resonance Frequency Analysis: A Novel Measurement Approach to Evaluate Acetabular Cup Stability During Surgery

**DOI:** 10.3390/s19224876

**Published:** 2019-11-08

**Authors:** Shunsuke Kikuchi, Katsuhiro Mikami, Daisuke Nakashima, Toshiyuki Kitamura, Noboru Hasegawa, Masaharu Nishikino, Arihiko Kanaji, Masaya Nakamura, Takeo Nagura

**Affiliations:** 1Department of Orthopaedic Surgery, Keio University School of Medicine, 35 Shinanomachi, Shinjuku-ku, Tokyo 160-8582, Japan; kikushun1111@gmail.com (S.K.); hikokanaji@gmail.com (A.K.); masa@keio.jp (M.N.); nagura@keio.jp (T.N.); 2Department of Biology-Oriented Science and Technology, Kindai University, 930 Nishi-Mitani, Kinokawa city, Wakayama 649-6493, Japan; kmikami@waka.kindai.ac.jp; 3Kansai Photon Science Institute, National Institutes for Quantum and Radiological Science and Technology, 817 Umemidai, Kizugawa city, Kyoto 619-0215, Japan; kitamura.toshiyuki@qst.go.jp (T.K.); hasegawa.noboru@qst.go.jp (N.H.); nishikino.masaharu@qst.go.jp (M.N.)

**Keywords:** total hip arthroplasty, implant stability, acetabular cup, laser, resonance frequency analysis, finite element method

## Abstract

Artificial joint acetabular cup stability is essential for successful total hip arthroplasty. However, a quantitative evaluation approach for clinical use is lacking. We developed a resonance frequency analysis (RFA) system involving a laser system that is fully contactless. This study aimed to investigate the usefulness of laser RFA for evaluating acetabular cup stability. First, the finite element method was performed to determine the vibration mode for analysis. Second, the acetabular cup was press-fitted into a reamed polyurethane cavity that replicated the human acetabular roof. The implanted acetabular cup was vibrated with pulse laser irradiation and the induced vibration was detected with a laser Doppler vibrometer. The time domain signal from the vibrometer was analyzed by fast Fourier transform to obtain the vibration frequency spectrum. After laser RFA, the pull-down force of the acetabular cup was measured as conventional implant fixation strength. The frequency of the first highest amplitude between 2 kHz and 6 kHz was considered as the resonance peak frequency, and its relationship with the pull-down force was assessed. The peak frequency could predict the pull-down force (*R*^2^ = 0.859, *p* < 0.000). Our findings suggest that laser RFA might be useful to measure acetabular cup stability during surgery.

## 1. Introduction

Shifting global demographics, predominantly an increase in an ageing population, has resulted in an increase in the number of patients with hip osteoarthritis worldwide [1]. Patients with severe hip osteoarthritis are often indicated for hip surgeries, such as total hip arthroplasty (THA). In Japan, approximately 50,000 patients undergo THA each year with an upward trend that is predicted to continue unabated [2]. Surgical outcomes have improved because of established surgical techniques and advances in implant design and materials [3]. However, fractures during surgery and loose implants during the early postoperative period have been reported as instances of complications whose prevalence has increased, particularly due to the increase in the number of surgeries among patients with fragile bones, which correspond to an increase in repetitive surgeries [4,5,6]. Therefore, novel approaches to prevent to THA surgery-related complications are warranted.

One such method is to achieve artificial acetabular cup stability. Insufficient acetabular cup stability can result in additional impaction or screw insertion, and these additional procedures can further lead to iatrogenic fracture or vascular injury [7]. Because there is no quantitative method to evaluate acetabular cup stability during surgery, this stability is currently measured through manual operator sensing. Although several quantitative methods have been established [8,9,10], they are not in clinical use. Others have evaluated acetabular cup stability according to torque [11] and pull-down force [12,13,14]. However, because the implants should be removed for measurement, their clinical utility is limited. Therefore, a quantitative method for evaluating acetabular cup stability without requiring the removal of the implant during the operation, thereby aiding in accurate surgery, should be established.

In dentistry, dental implant stability is evaluated by resonance frequency analysis (RFA) [15,16,17,18,19], based on the theory that the resonance frequency obtained from dental implants vibrated using a magnetic force generated by small magnet (smart peg) and the vibration waveform detected can be analyzed. Because dental implants are exposed in the oral cavity, it is convenient to adapt RFA for dental implants [20]. Conversely, adaptation of RFA for an acetabular cup is difficult because it is impacted into the pelvis and is located deep within the human body. For measuring acetabular cup stability, hammer tapping [10,21], and implant vibration [8,22,23,24] techniques have been used. Hammer tapping is subjective, is dependent on the surgeons, and is difficult to tap with the same power. The stability of the vibrator, examined by only manual sensing, is indicated as either stable or unstable. However, the unwieldy and complicated nature of these approaches renders them impractical for clinical use as quantitative methods.

To overcome these limitations, we developed a contactless RFA vibrometer based on pulse laser and laser Doppler that does not involve the direct attachment of devices, such as magnets, to the implant. Laser RFA has been used as one of the laser remote-sensing approach for inspecting internal defects within concrete structures, such as tunnels [25,26]. The basic principle of laser RFA is irradiation of the sample by pulse laser to induce surface vibration via laser ablation, which is detected using a laser Doppler vibrometer. The scheme of laser RFA is similar to that of hammer inspection which provides rapid, contactless, and quantitative measurements. Further, due to its remote diagnostic scheme, laser RFA can be performed safely. The aim of the present study was to determine whether peak frequency of laser RFA was a predictor of the pull-down force. To achieve this goal, we used finite element method (FEM) analysis as a preliminary approach to determine the appropriate vibration mode and then examined laser RFA.

## 2. Materials and Methods

### 2.1. Finite Element Method to Analyse the Vibration Mode

The properties of acetabular cup and polyethylene foam are presented in Table 1. The FEM model was constructed by combining hemisphere models simulating the acetabular cup and high-density polyethylene foam (Figure 1). Our FEM model was a modal 3D model and 3-mm mesh model made of the titanium alloy Ti-6Al-4V, a typical biocompatible material exhibiting high stiffness. The acetabular cup was made of Ti-6Al-4V, whereas the polyethylene foam was made of simple polymers, compared to the polyethylene foam containing carbonyl and amino groups.

Finite element analysis was performed using Fusinon360^®^ (Autodesk Incorporated, San Rafael, CA, USA). In our study, there was a gap between the polar area of the acetabular cup and bottom of the reamed cavity when the acetabular cup was fixed [27] (Figure 2a). To reproduce this fixation, five different coverage areas were calculated as clinical imitation models. The coverage area was adjusted according to the height of the polyethylene foam (5 mm, 10 mm, 15 mm, 20 mm, and 30 mm. 30 mm indicates full coverage and 20 mm indicates that 5 mm from the polar area of the acetabular cup was not covered) (Figure 2b–f). For the FEM model, it was a modal FEM analysis, with all side surfaces of the polyethylene foam being kinematically constrained and the contact area between the cup and the polyethylene foam being unified with adhesion. Friction and stiffening effects were difficult to incorporate in the FEM model. However, the area of the contact surface indicated acetabular cup stability. The relationship between coverage area and peak frequency was evaluated using Pearson’s correlation coefficient.

### 2.2. Material

Four blocks of polyurethane foam (SAWBONES^®^, Pacific Research Laboratory, Inc., Vashon Island, WA, USA; SKU: 1522-01, 03 and 04; 0.48 g/cm^3^) were used to represent human acetabular bone. The acetabular cup SQRUM^®^ HA non-hole (Kyocera Medical Corporation, Osaka, Japan; 50-mm diameter) was used.

### 2.3. Measurement

An orthopaedic surgeon (SK) with >five years of experience in hip surgery performed all procedures to fix the acetabular cup to the polyurethane foam. All examinations were confirmed by two additional orthopaedic specialist surgeons with 12 and 28 years’ experience. Methods were discussed between specialists and the results were applicable to all surgeons.

A bone cavity that replicated the acetabular cup was made by standard operative techniques. For adequate press-fit, a 48-mm reamer (Kyocera Medical Corporation) was used. The acetabular cup connected to a 310-mm rod was impacted into this cavity with the tapping technique using a hammer (Figure 3a). After disconnecting the rod, vibration was induced in the implant with an impact laser (Nd:YFL laser Model 1050; JP Innovation, LLC, Monroe, WA, USA), and the wavelength and pulse width were 1053 nm and 10 ns, respectively. The frequency was detected with the laser PDV-100 (Polytec Inc., Irvine, CA, USA) (Figure 4a). The rod was connected again and the pull-down force (N) was measured using the force gauge DPX-10TR (IMADA Co., Ltd., Aichi, Japan) (Figure 3b). We reamed one cavity in each block and prepared four blocks. These experiments were repeated thrice for each polyurethane foam block. Thus, a total of 12 experiments were performed.

### 2.4. Laser Resonance Frequency Analysis

The experimental layout of the laser system is shown in Figure 4b. First, the laser spot size of the impact laser pulse was adjusted by a beam expander, followed by adjustment of the beam, and focused with a focusing lens. The beam diameters of the focusing spot were 0.65 mm (horizontal) and 0.50 mm (vertical), and these values were defined as the 1/e^2^ values from the peak values of the Gaussian fitting for the bean profile (Figure 4c). The irradiation pulse energy was set to 46 mJ, and it was evaluated with an energy meter (QE25LP-H-MB-QED; Gentec Electro-Optics, Inc., Québec, QC, Canada). The pulse laser system was operated at a repetition rate of 10 Hz. The laser Doppler vibrometer was continuously irradiated to detect the induced vibration. The irradiation points between the pulse laser and laser Doppler vibrometer had displacement of 1 mm to inhibit external noises, such as a laser ablation plume and plasma emission. The measured signal from the laser Doppler vibrometer was acquired using a multi-function measuring system (RioNote; RION Co., Ltd., Tokyo, Japan) that was synchronized with the timing of laser pulse irradiation (Q-switch operation signal), and saved for 16 s, i.e., data of 160 pulse irradiations. The signal obtained included laser-induced vibrations with 160 laser pulses that were divided according to pulse irradiation and averaged with all divided data. The correlation function of each divided signal with the averaged signal was evaluated, and the top 20 signal data with regard to the correlation function were extracted to eliminate singularity. The extracted signal data were averaged for analysis. Data until 2 ms after laser irradiation were purged to obtain a clear signal of the fundamental and low-order characteristic vibration patterns (CVs) in the audible area, as a short impact period, i.e., laser pulse width (10 ns), has the potential to induce a wide spectrum of vibrations until ultrasonic vibration (~MHz). Finally, the purged data were analyzed to obtain the frequency spectrum by fast Fourier transform (FFT) with a rectangular window function.

### 2.5. Statistical Analysis

The relationship between peak frequency (Hz) and pull-down force (N) was examined by power approximation in order to predict the pull-down force from peak frequency. All statistical analyses were performed using SPSS Statistics software version 25 (International Business Machines Corporation, Armonk, NY, USA).

## 3. Results

### 3.1. Analysis of the Vibration Pattern Using Finite Element Method

Five vibration patterns (types A–E) were identified (Figure 5a). Vibration of the polyethylene foam (type A) reflected bone quality. Types B and D were shaking patterns that reflected friction. Types C and E were mixtures of expansion and contraction that reflected engagement surrounding the outer periphery. The peak frequency of laser RFA was hypothesized to be the predictor of pull-down force, which is a measure of conventional stability that was mostly reflected in the type B pattern.

The measurement of peak frequency in all bands showed a strong and significant correlation between coverage area and peak frequencies in all the vibration patterns (type B, *R* = 0.981; type C, *R* = 0.923; type D, *R* = 0.941; type E, *R* = 0.927) (*p* = 0.001) (Figure 5b). Measurement bands were not limited, similar to experiments with actual laser devices. This result suggested that high acetabular cup stability was associated with high peak frequency, which was consistent with the findings of the latter RFA experiment.

### 3.2. Laser Resonance Frequency Analysis

At each pull-down force, the frequency spectra obtained by the FFT of time domain waveforms were evaluated 12 times. There was no breakage of the polyurethane foam before and after the experiments. Articulate frequency >10 kHz was absent in the analysis of the vibration signal after laser irradiation for 2 ms, which was sufficiently attenuated for high-order CVs in the frequency region. The frequency of the first highest amplitude between 2 and 6 kHz was considered the peak frequency, assumed as type B based on the FEM method, which correspondingly increased with the pull-down force (Figure 6a). Therefore, analysis of the highest peak at a pull-down force of 3.82 N was difficult.

Regression analysis (F (1, 10) = 61.149) showed a significant correlation (*R*^2^ = 0.859, *p* < 0.000). These results indicated that the peak frequency was a significant predictor of the pull-down force (Figure 6b), which was predicted as 2.062 × 10^−85^ (peak frequency)^25.650^. The frequency resolution of obtained data was 3 Hz. Therefore, the horizontal axis error bar in Figure 6a is assumed ±3 Hz. 6 Hz error to 2000 Hz is 3 × 10^−3^. Therefore, considering the ratio 25.65, the relative error is approximately 7.7% (±3.85%) in the measurement frequency band.

## 4. Discussion

To the best of our knowledge, this is the first study to demonstrate that laser RFA successfully predicted the pull-down force of the acetabular cup using a simplified bone surrogate model that indicated the possibility of using laser RFA to evaluate acetabular cup stability. There are some differences and improvements compared with its application in other fields. Remoteness for measuring several meters away is unnecessary because it can be operated at hand. Alternatively, the lower irradiation pulse energy is appealing for surgical operation. Our evaluation is achieved by approximately 1/100 of pulse energy compared with its application in concrete inspection system whose irradiation pulse energy is 1000–4000 mJ [25,26]. If we can achieve optimization of the spot size by developing a designated device, it will lead to higher fluence. The higher fluence enables us to acquire signals more efficiently as observed in the previous study [26].

Acetabular cup stability can be evaluated using a vibrator with RFA in a cadaver model [8] or a hammer tapping instrument using model bones [10]. Although both methods can reasonably measure acetabular cup stability, they are limited in utility due to several reasons. Measurement of pull-down force during surgery is destructive because it leads to the detachment of the acetabular cup from the pelvic bone. Although hammer tapping is non-destructive, the tapping force depends on the operators, leading to poor repeatability and reproducibility.

The advantage with RFA is that the vibration induced by a magnetic force is reproducible and non-destructive and not subjected to the operator’s hand movements. Compared with dental implants, the acetabular cup is located deep within the human body, rendering it difficult to attach the ‘smart peg’ for RFA. Debruyne et al. used the Osstell^®^ apparatus for inducing vibration with magnetic force and used laser for detecting vibration [28]. To the best of our knowledge, this is the first study to perform RFA using a laser for both vibration excitation and detecting vibration.

Comparisons of conventional methods are presented in Table 2. Relative to other stability evaluation systems, laser RFA offers several advantages. First, laser RFA is a contactless system that can rapidly determine acetabular cup stability, with only 10 s from laser irradiation to frequency spectrum analysis. By contrast, hammer tapping is more complex and requires 16 impacts to obtain sufficient data for analysis. Second, laser RFA can repeatedly irradiate a narrow area—an essential property to evaluate cup stability—particularly during surgeries, such as THA, that are performed in a limited working space in the body.

As mentioned above, the impact of the hammer instrument varies considerably [9,21], whereas laser consistently emits the same energy with each irradiation and results in precise data by averaging. A previous study showed that the frequency spectrum obtained from hammer tapping of the acetabular cup did not reveal the relationship between fixation strength and polar gap or the relationship between polar gap and frequency shift [29]. These limitations prevent a comprehensive analysis of implant stability and the experimental results in the previous study require further research.

Compared with the vibration pattern of screw-shaped dental implants, the pattern of the acetabular cup is complex because of its hemispherical shape, outer periphery, and friction between the bone and acetabular cup surface. Higher coverage area was associated with a higher peak frequency in our study. Higher bone contact area was previously associated with greater acetabular cup stability [14], which is consistent with our results.

The mechanism behind acetabular cup fixation is the engagement surrounding the outer periphery [30]. The edge form and elasticity of bone are important factors that alter laser-induced vibration. Friction between the bone and acetabular cup surface is another mechanism. A conventional FEM study using composite bone and an acetabular cup (Ti-6Al-4V) showed >10 vibration patterns, and acetabular cup fixation was assessed by modal shape curvature [31]. In that study, one pattern had a torsional characteristic and the remaining nine had a combination of bending and torsional characteristics. In our study, five vibration patterns were detected, of which four were related to acetabular cup stability. Two patterns were assumed to mainly reflect engagement surrounding the outer periphery, whereas the remaining two patterns were assumed to mainly reflect friction between the acetabular cup surface and polyurethane foam. To determine the peak resonance frequency, it was important to identify the pattern that reflects the pull-down force the most. Because several peak frequencies in the frequency spectrum positively correlated to the pull-down force, it was unfeasible to determine the nominal peak resonance frequency for stability analysis.

The limitation of FEM analysis was the lack of typical FEM models for calculating friction. Due to the complexity of incorporating friction in the FEM model, we hypothesized that the contact area between the acetabular cup and bone leads to friction, thereby reflecting the acetabular cup stability, which in turn affects its natural frequency. Therefore, high acetabular cup stability was associated with high peak frequency, which is consistent with the findings of the RFA experiment.

The press-fit model was difficult to express using the FEM model, and under-reaming was not considered. Laser RFA vibrates the acetabular cup rather than the mounting base. Therefore, the boundary surface should be considered. The effect of the coverage area is sufficient to explain the tendency of the vibration patterns because the coverage area reflects pre-stress. Although pre-stress does not induce the transformation of the acetabular cup, in reality, it is essential to set the interface model considering the residual stress of the bone and non-linear response of the soft tissue.

Our preliminary study verified whether the laser RFA principle was applicable without using the non-linear response due to the use of polyurethane foam. We showed that the vibration was type B, whereas type C was the shift in vibration from 3000 Hz to 5000 Hz, which was based on our vibration frequency region from the results of the FEM analysis. These findings were then extrapolated to a clinical study using cadavers.

In the present study, we selected the pull-down force as the stability force, although pull-down force is not a typical method for comparing pull-out and torque forces. Because our study was based on previous findings [14], we excluded force displacement curve and attempted to establish pull-down force as an ASTM standard method. Polyurethane foams are widely used as standard test materials for mimicking the human cancellous bone [32]. The mechanical properties of the human bone and polyurethane foam are shown in Appendix A
Table A1. The dimension of the foam and the adopted value of under-reaming were not determined because a previous study [10] did not show the value of under-reaming. Vibration induced by laser is focal and the amplitude is <1 μm, and the vibration does not lead to the polyurethane foam block. Clinically, there was no mechanical damage to the bone, showing similar correlations that are independent of the mounting base. Re-using the foam block leads to different results because it was assumed that friction reduces with decrease in the pull-down force. We recorded pull-down force for each insertion and the effect of re-use. Re-tapping is occasionally performed during surgeries because of mal-installation, therefore, the re-use of polyurethane foam reflects clinical situations. It is plausible that the results vary with the bone type and future studies will examine the pelvis bone.

The acetabular cup was not subjected to heat because the examination was performed at room temperature reflecting the clinical situation, and the temperature of the acetabular cup was not measured. The nanosecond pulse width of the impact laser may not increase the temperature of acetabular cup because the specific heat of the titanium material Ti-6Al-4V is 610 mJ/g K and the input energy was only 46 mJ, which is a small energy pulse.

The debris generated by vibration associated with laser is a significant concern because there could be an adverse reaction to metal debris [33,34,35,36,37]. A possible solution to this is the design of an external device connected to the acetabular cup. The external device will be cannulated and connected to an insertion rod that will preserve repeatability and reproducibility, because the debris will remain inside the external device and not inside the acetabular cup, thereby preventing adverse effects associated with debris during the use of such a device. Another concern is the use of reflection by fixed mirrors in the current laser system, reducing flexibility in the irradiation range. It is important to place the acetabular cup in the right position for irradiation. Therefore, the body position should be changed vertically and horizontally to evaluate acetabular cup stability. Because free body movement is not possible, assessment will be difficult in the operation room. However, the use of an optical fibre system can help overcome this issue owing to its flexibility. An optical fibre system enables the operator to irradiate an arbitrary point, and this will be a future in vivo study, prior to which, using a cadaver, we will establish the use of an optical fibre system and confirm safety.

The incidence of loosening that could be prevented with this device is indeed low, and complications, such as infection and embolism, cannot be prevented with this device. However, this system enables orthopaedic surgeons to prevent iatrogenic pelvic fracture induced by excessive tapping or vascular injury induced by additional screw insertion, which can be fatal. This system has the potential for use in revision surgery, where often the surgeon is required to decide whether to preserve the existing cup or to replace it. We predict that our system might prevent morbidities caused by the unnecessary removal of the existing cup or by retaining a cup with insufficient fixation in its position.

In conclusion, our findings suggest that laser RFA might be a useful approach to measure acetabular cup stability during surgery.

## Figures and Tables

**Figure 1 sensors-19-04876-f001:**
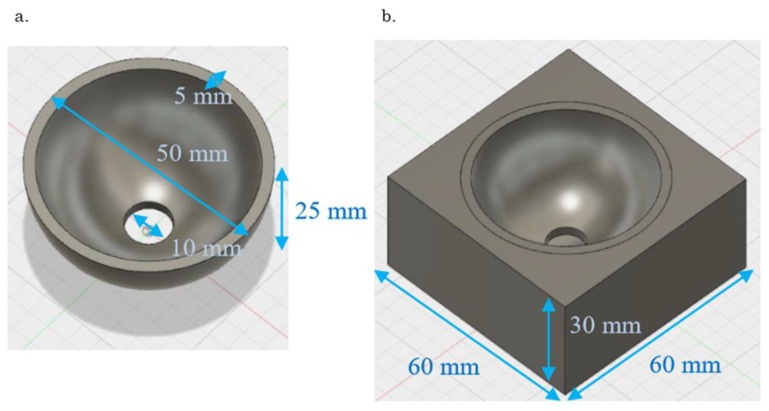
Finite element method model (**a**) Acetabular cup with a titanium-based alloy, (**b**) Acetabular cup with polyethylene foam.

**Figure 2 sensors-19-04876-f002:**
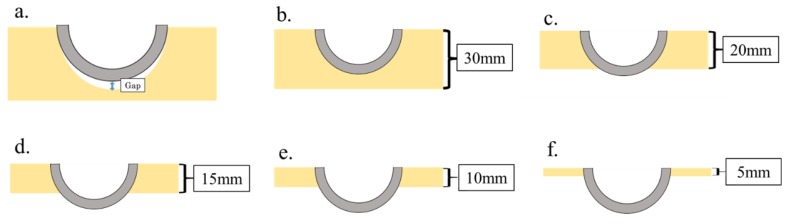
(**a**) Fixation image in clinical situations. There is a gap between the polar area of the acetabular cup and the bottom of the reamed cavity; (**b**–**f**) Clinical imitation models. Coverage was defined by the height of the polyethylene foam: (**b**) 30 mm, (**c**) 20 mm, (**d**) 15 mm, (**e**) 10 mm, and (**f**) 5 mm.

**Figure 3 sensors-19-04876-f003:**
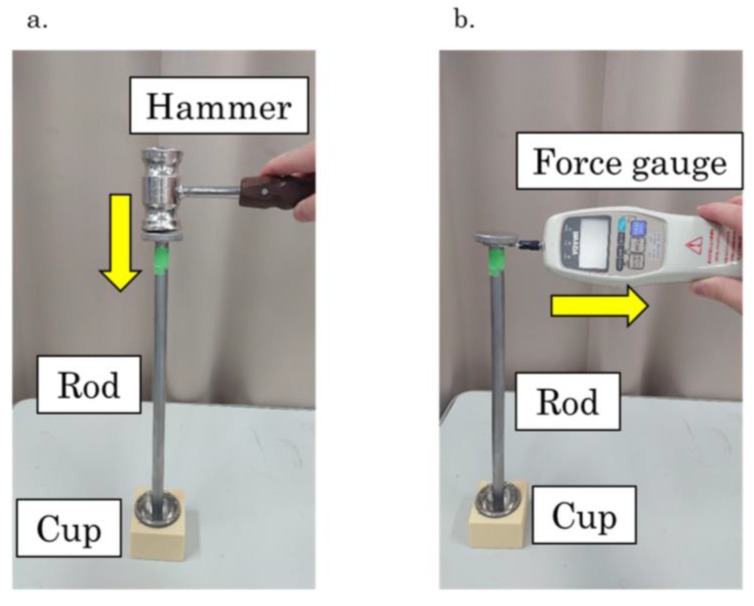
Evaluation schemes for the acetabular cup (**a**) mounting to polyurethane form and (**b**) the evaluation of pull-down force defined as the mounting stability.

**Figure 4 sensors-19-04876-f004:**
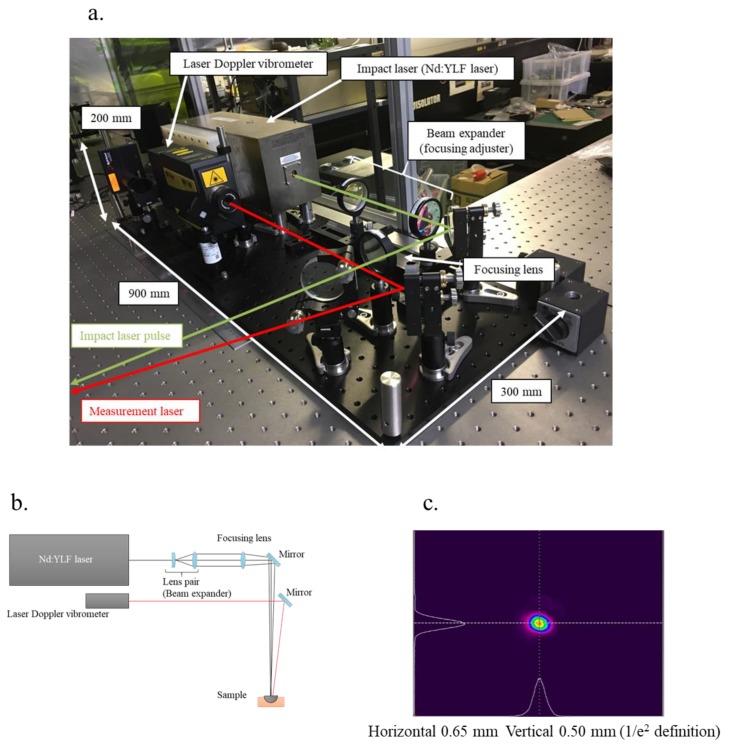
(**a**) Image of the laser system with a pulse laser and a laser Doppler vibrometer, experimental details of (**b**) the layout of the laser system, and (**c**) the beam profile of the Nd:YLF laser at a focus point, i.e., irradiation pattern for the acetabular cup.

**Figure 5 sensors-19-04876-f005:**
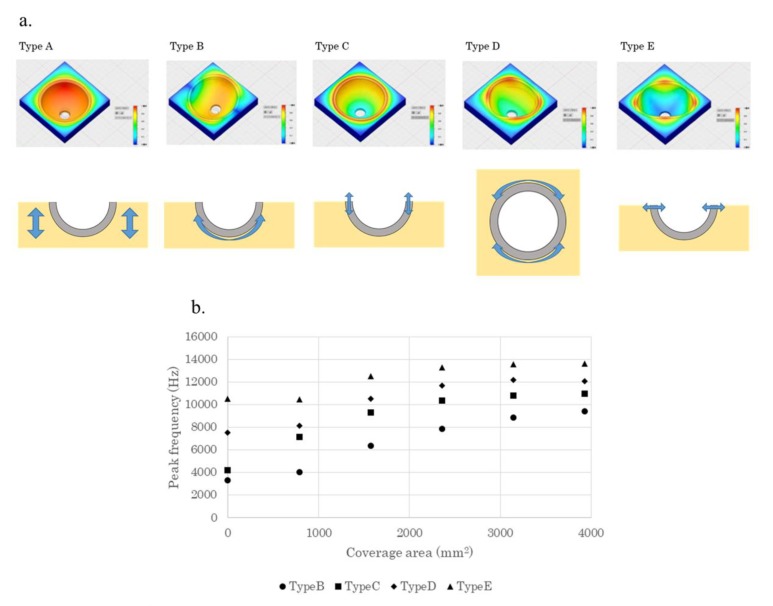
(**a**) Five patterns of vibration detected in this study. Type A is vibration of the polyethylene foam; type B is shaking of the acetabular cup body; type C is expansion and contraction of the acetabular cup body; type D is shaking of the acetabular cup rim and type E is expansion and contraction of the acetabular cup rim; (**b**) The relationship between the coverage area and peak frequency. The peak frequency correlated strongly with the coverage area with regard to the patterns.

**Figure 6 sensors-19-04876-f006:**
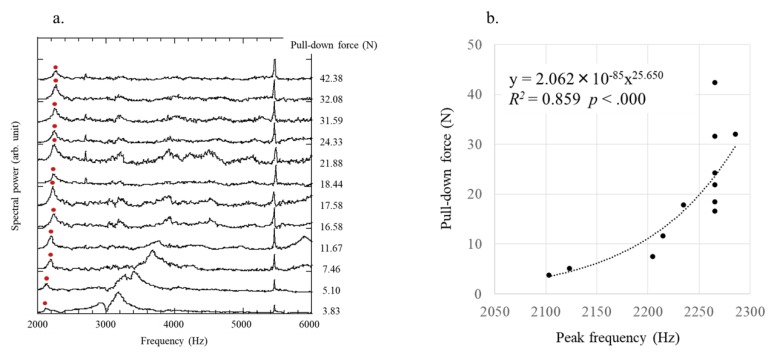
(**a**) Frequency spectra of laser-induced vibration of the acetabular cup for each pull-down force. The frequency of the first highest amplitude was defined as the peak frequency, as shown by the red circle; (**b**) The relationship between the pull-down force and peak frequency shows a strong correlation between the two parameters.

**Table 1 sensors-19-04876-t001:** Material properties of acetabular cup and polyethylene foam.

	Acetabular Cup (Ti-6Al-4V)	Polyethylene Foam
Thermal Conductivity (W/m·k)	6.7	0.211
Specific Heat (J/g·k)	0.526	2.859
Thermal Expansion Meter Rate (μm/m·k)	8.600	150.00
Young’s Modulus (GPa)	113.770	0.911
Poisson Ratio	0.34	0.39
Modulus of Shearing Elasticity (MPa)	42,451.493	320.000
Density (g/cm^3^)	4.43	0.952
Yield Strength (MPa)	882.529	20.670
Tensile Strength (MPa)	951.477	13.780

**Table 2 sensors-19-04876-t002:** Comparisons of conventional methods: laser resonance frequency analysis has the advantages of protectability, repeatability, reproducibility, and contactability.

	Protectability	Repeatability	Reproducibility	Contactability
Pull-Down	×	×	×	×
Hammer Instrument	○	×	×	×
Dental RFA	○	○	○	×
Laser RFA	○	○	○	○

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
