# Peer review of "Laser Resonance Frequency Analysis: A Novel Measurement Approach to Evaluate Acetabular Cup Stability During Surgery"

_sensors, 2019, doi:10.3390/s19224876_

Round 1
Reviewer 1 Report
The authors present in this paper the use of laser resonance frequency analysis (RFA) to evaluate the stability of acetabular cup. This is an innovative and attractive work because it studies to perform RFA for both vibration excitation and detectin vibration. The presentation of the work in its different subsections is correct and clear. And the experimental results are presented meticulously. However, the final result of the relationship between the resonance frequency and the pull-down force applied must be studied/explained in more depth.
The exponential of this ratio is around 25, so an uncertainty propagation estimates that the relative error found in the value of the force will be 25.6 times greater than the relative error in the estimation of the frequency. How accurate is the estimation/measure of this frequency? What errors will be made in the estimation of the force? Are these errors compatible with their use in the clinical applications exposed?
Reviewer 2 Report
In the paper, the author introduces a method of laser resonance frequency analysis to evaluate acetabular cup stability during surgery. The method is relatively new in the medical field. However, I still have several questions for the authors:
This method is widely used in other fields. Could the authors explain the difference and improvement of this method when it is used in the medical field? It is shown that the size of the focusing spot is 0.65 mm and 0.5 mm, quite large. In practical, the size can be much smaller by choosing a suitable optical design. Could the authors provide some explanation on this point? Figure 6b shows the relation between peak frequency and pull-down force. However, the fitting precision seems low. I recommend the authors to do reprocessing for the data and then draw conclusions.Author Response
Please see the attachment.

Round 2
Reviewer 1 Report
The authors' responses to the comments and indications made during the review process are satisfactory.
Reviewer 2 Report
The manuscript has been significantly improved. I suggest to publish it.